# Acute Effects of 30 g Cyclodextrin Intake during CrossFit^®^ Training on Performance and Fatigue

**DOI:** 10.3390/jfmk9010027

**Published:** 2024-01-30

**Authors:** Franscisco Javier Grijota, Víctor Toro-Román, Ignacio Bartolomé, Elías Cordero-Román, Cristian Sánchez López, Jose Miguel Jiménez, Ismael Martínez-Guardado

**Affiliations:** 1Sport Sciences Faculty, University of Extremadura, Avenida de la Universidad s/n, 10003 Cáceres, Spain or franciscojavier.grijota@ui1.es (F.J.G.); ignbs.1991@gmail.com (I.B.); 2Faculty of Humanities and Social Sciences, Isabel I University, C. de Fernán González, 76, 09003 Burgos, Spain; 3Department of Health Sciences, Research Group in Technology Applied to High Performance and Health, TecnoCampus, Universitat Pompeu Fabra, 08302 Barcelona, Spain; 4Education Faculty, Pontifical University of Salamanca, Henry Collet Street, 52-70, 37007 Salamanca, Spain; 5Elías Nutrition, 10003 Cáceres, Spain; eliascorderoroman@gmail.com; 6Try It Studio, C. Gonzalo Mingo, 3, 10004 Cáceres, Spain; cristian@try-it.es (C.S.L.); josemi@try-it.es (J.M.J.); 7Faculty of Health Sciences, Camilo José Cela University, C. Castillo de Alarcón, 49, Villafranca del Castillo, 28692 Madrid, Spain

**Keywords:** CrossFit, performance, supplementation, highly branched cyclic dextrin, clusterdextrine

## Abstract

The main objective of this study was to investigate the influence of carbohydrate intake (cyclodextrin) on performance during the performance of two consecutive workouts of the day (WODs) lasting 20 min each. Twenty-one male CrossFit (CF) athletes (29.5 ± 4.3 years; 72.81 ± 12.85 kg; 1.74 ± 0.06 m; 3.41 ± 1.21 years of experiences) participated in a crossover, randomized, and double-blind study. The effect of supplementation with 30 g of cyclodextrin (SG) (Cluster Dextrin^®^) or placebo (PG) (Bolero Advanced Hydration^®^) was evaluated on the performance of two specific WOD. Additionally, the effect on handgrip maximum strength, countermovement jump (CMJ), Wingate test, and 1 RM bench press test was evaluated. The effect on blood glucose and lactate was also evaluated. No differences were found in time, height, and power (W/Kg) in CMJ. However, there was a percentage improvement in CMJ jump power (W) (*p* < 0.05) between the groups, assuming an improvement in performance due to the intervention. Moreover, both conditions experimented differences in execution speed between sets (*p* < 0.05) in pre-WOD, and differences in post-WOD only in the placebo group, as well as decreases in this variable per repetition across the set (*p* < 0.01) in both conditions. However, no differences were found in the rest of the variables. Supplementation with 30 g of cyclodextrin did not have any metabolic or performance effects in CF tests. Although some differences between groups were observed in CMJ and power tests for bench press, the data are not conclusive and further research is needed in this regard.

## 1. Introduction

CrossFit^®^ (CF) is a sports modality that is on the rise due to the popularity it has gained over the past 10 years. Due to its varied functional movements, performed at a relatively high intensity through metabolic conditioning, physical exercise, and weightlifting, this modality has been gaining popularity around the world [1]. There are different types of CF exercises, and one of the most common is the workout of the day (WOD). It is considered a concurrent training method, where strength and endurance exercises are combined in short, high-intensity daily sessions with a marked interval character [2]. The purpose of a WOD is to achieve the best possible time or the highest number of rounds and repetitions within a given time domain [1].

This type of intermittent activities are characterized by a high metabolic stress, where anaerobic power and aerobic capacity are of great importance [3]. Previous authors reported average intensities of approximately 90% of the maximum heart rate (HRmax) and aerobic intensities with an average of approximately 85% of the maximum oxygen consumption (VO_2_max) [4]. However, due to the intermittent nature of the efforts and the high intensities, from a physiological point of view, CF is very similar to high-intensity interval training (HIIT), with the particularity of performing numerous strength and muscular endurance exercises [5]. It has been shown that in this type of intermittent task, muscle glycogen is one of the main limiting factors of performance, as well as that this type of task is the one that generates the greatest depletions of it [6].

The growing popularity of CF-based exercise methods indicates that these methods are becoming increasingly common among athletes looking to improve their performance. Currently, nutritional habits and the use of nutritional supplements have already begun to be studied [7]. It is interesting that, despite the current knowledge about the importance of pre-exercise glycogen levels in this type of sport, CF practitioners develop different dietary models that can compromise these levels, such as intermittent fasting, paleo diets, or ketogenic diets, among others [8].

With respect to the use of nutritional supplements, existing data show that a wide variety of substances are consumed, some of them related to energy production and carbohydrate (CHO) intake [7]. Recent research has observed beneficial effects of CHO intake on short-duration exercises (<25 min) [9] and in intermittent exercises (>60 min) [10].

A training of an intense nature, such as CF, implies a high and continuous demand for glycolytic energy production, increasing the use of glycogen, both hepatic and muscular [11]. Given the above, it is possible that an inadequate intake of CHO during a CF training period could compromise glycogen replenishment and performance in subsequent workouts and competitions [12]. It has been shown that the intake of carbohydrates (CHO) during resistance exercise delays neuromuscular fatigue and significantly improves exercise capacity and work rate with a dose–response relationship [13,14]. In addition, it is well known that the availability of CHO substrates plays a central role in peripheral fatigue, but also within the central nervous system and, therefore, in central fatigue.

One of the main and most innovative types of CHO marketed in the field of sports nutrition today is highly branched cyclic dextrin (HBCD), commonly referred to as cyclodextrin [15]. HBCD is a CHO derived from plant starch, composed of D-glucose molecules forming highly branched structures [16]. These molecules are characterized by having high rates of digestibility and absorption in humans, with the D-glucose molecules becoming bioavailable in the bloodstream relatively quickly. It has been reported in the literature that HBCD has highly beneficial properties for athletes, such as adequate gastric emptying and a favorable glycemic response [16]. Like cyclodextrin, after the ingestion of 15 g of HBCD, a considerable increase in blood glucose levels is observed after 15 min of ingestion, with HBCD presenting a slightly higher response than cyclodextrin [15]. Another interesting property of HBCD compared to other commercial CHO formulations is that, given its low osmolarity and high molecular weight, it presents better gastric emptying processes in humans, which leads to faster and less aggressive assimilation, as well as a better and more accelerated bioavailability of D-glucose molecules in the body [16]. Moreover, it is considered a CHO of the latest generation, and it has been observed that it presents interesting properties for athletes, such as a better gastric emptying or a better glycemic response than other similar products [15]. However, despite its properties and its massive marketing, it is a product that is poorly researched at present, and the effects of supplementation with cyclodextrin during a CF workout are very scarce, if not non-existent.

We hypothesized that the ingestion of HBCD (cyclodextrin) would result in a smaller decrease in performance after CF training. Therefore, the main purpose of this study was to investigate the influence of the intake of HBCD (cyclodextrin) on performance during the performance of two consecutive WODs lasting 20 min each.

## 2. Materials and Methods

### 2.1. Participants

A total of twenty-one male CF athletes participated in the study (Table 1). All participants were informed about the purpose of the study and signed a consent form before enrolling. Participants maintained their training throughout the study (~21 days). To participate in the study, the following inclusion criteria were established: (a) more than 2 years of experience in CF; (b) regional, national, and/or international competition level; (c) no cardiovascular, respiratory, metabolic, neurological, orthopedic diseases or disorders that could affect the performance of the test; (d) no consumption of drugs or medications; (e) no smoking; (f) no consumption of sports supplements or pharmacological products during the study or in the 6 months prior to the start of the study. The protocol was reviewed and approved by the University of Extremadura bioethics committee (176/2023) following the guidelines of the Declaration of Helsinki, updated at the World Medical Association Assembly in Fortaleza (2013) for research on human subjects. Each participant was assigned a code to maintain their anonymity.

### 2.2. Design

The present study had a randomized, double-blind, crossover design (Figure 1). The procedures applied in the study lasted three weeks, during which four visits were made to the CF center (Figure 2).

In week 1, the participants attended the CF training center twice. On day 1 of week 1, a researcher explained all experimental procedures and the subjects signed informed consent forms. On day 2 of week 1, the participants performed all assessment tests so that the subjects could become familiar with them. These tests were a manual dynamometry test, CMJ vertical jump test, bench press speed test, and a Wingate test on a cycle ergometer, performed in that order. Afterwards, the first placebo (GP) and supplemented (SP) groups were established at random using a specific software (https://www.randomizer.org, version 4.0, accessed on 1 October 2023). The day before the assessments of week 2 and 3, anthropometric, body composition, and nutritional assessment evaluations were performed.

In weeks 2 and 3, the participants performed the fitness assessments and the WOD in different study groups each week (SG or PG). All subjects performed the assessments on the same day and at the same time. Both sessions were always conducted under similar environmental conditions (temperature 20–24 °C, relative humidity 45–55%). In the two days prior to the assessments in week 2 and 3, the athletes did not perform CF training. On the day of assessment, the participants first performed the fitness assessment tests in the order established in Figure 2: (1) hand grip strength; (2) CMJ; (3) bench press speed; and (4) Wingate test. The tests were performed following that order, and a passive rest of 5 min was established between them.

After the fitness assessments, the athletes had a 30 min rest before performing the WOD. After completing the first WOD, the athletes rested for 20 min during which, after the first 2 min of recovery, they ingested a beverage with HBCD (SG) and another with water and dye (PG). The SG intake consisted of 30 g of the product (described later) diluted to 8% in 375 mL of mineral water. This intake was performed in a maximum of 5 min.

After 20 min of rest, the athletes performed the same WOD again. During both WODs, repetitions were counted, and the average and maximum heart rate were monitored. Three minutes after completing the second WOD, the athletes returned to perform the different assessments in the order established above.

For more comprehensive monitoring, lactate and glucose assessments were performed at different times during the study (Figure 2).

### 2.3. Diet and HBCD Consumption

A nutrition professional established nutritional guidelines to ensure that all study subjects followed a similar diet 48 h before the start of testing, which consisted of ~60% CHO (5.5 g CHO per kg of body weight), 25% lipids, and 15% proteins. The diet was recorded by the participants 24 h before the first experimental test (week 2), and the same diet was replicated 24 h before the second test (week 3). Compliance with the established dietary instructions was assessed by checking the participants’ records through the “My Fitness Pal” app [17].

Participants were asked to avoid caffeine, alcohol, and other supplements to avoid any interaction with supplementation. Similarly, the intake of coffee, tea, and energy drinks containing caffeine was completely avoided. Participants were alerted to possible side effects such as gastrointestinal symptoms. No participant was under pharmacological treatment, nor did they present any symptoms when ingesting the drinks.

The intake of cyclodextrin or placebo occurred at the end of the first WOD and 10 min before the second WOD started. The HBCD (Glyco Proe-Cyclicdextrin (Cluster Dextrin^®^)) consumed in the present study by SP presented the following characteristics per 100 g of product: energy value: 1649 kJ/388 kcal; fat: 0 g; of which saturated: 0 g; CHO: 97 g; of which sugars: 2 g; protein: 0 g; salt: 0 g.

In contrast, the drink consumed by the PG group (Bolero Advanced Hydration^®^) presented the following nutritional information per 100 mL of product: energy value: 7.32 kJ/1.71 kcal; fat: 0 g; of which saturated: 0 g; CHO: 0 g; proteins: 0 g; salt: 0.0025 g; vitamin C: 6 mg.

Both drinks were supplied in unlabeled 500 mL bottles of a garnet red color to avoid athlete interpretation. The amount of water in the bottles was 375 mL to respect an 8% product dissolution [18].

### 2.4. Anthropometric Measurements

The morphological characteristics of the participants were evaluated in the morning and under identical conditions (fasting, naked, and barefoot). All measurements were performed by an experienced researcher in cineanthropometric techniques, in accordance with the recommendations of the Society for the Advancement of Cineanthropometry [19]. The anthropometry and body composition test were performed the day before each assessment. Height was measured with a wall stadiometer with a precision of 0.1 cm (Seca 220, Hamburg, Germany).

Weight was measured using a digital scale with a precision of 0.01 kg (Seca 769, Hamburg, Germany). The following instruments were used for anthropometric evaluations: a Holtain^©^ 610ND skinfold caliper (Holtain, Crymych, UK), with a precision of ±0.2 mm; a Holtain^©^ 604 bone diameter caliper (Holtain, Crymych, UK) with a precision of ±1 mm; and a Seca^©^ 201 tape measure (Seca, Hamburg, Germany) with a precision of ±1 mm. To calculate the percentages of muscle mass and body fat, the equations of the Spanish Cineanthropometry Group were used [20].

The anthropometric measurements obtained were height, weight, 6 skinfolds (abdominal, suprailiac, subscapular, triceps, thigh, and leg), bone diameters (bistyloidal, biepicondylar humerus, and biepicondylar femur) and muscle perimeters (relaxed arm and leg).

### 2.5. Dynamometry Test

Grip strength was measured using a Takei 5101 dynamometer (Takei Instruments Ltd., Tokyo, Japan). Participants performed two maximal voluntary contractions with the dominant hand and arm fully extended. The dynamometer grip was adjusted to the participants’ hands individually and following the manufacturer’s guidelines [21]. Two attempts were made and the best value was selected for further analysis.

### 2.6. Jump Test (CMJ)

Lower body explosive strength was assessed via a countermovement jump (CMJ) test [22]. The previous test was chosen to measure the neuromuscular function of the leg extensor muscles given that they can achieve it with a high degree of reliability [23]. A jump mat system (Chronojump Boscosystems, Barcelona, Spain) was used to measure the flight time and jump height. Three attempts were made with a 30 s rest between jumps and the best jump was selected for further analysis. All jump tests were performed following the guidelines of Markovic et al. [24] and Rodríguez-Rosell et al. [23].

### 2.7. Wingate Test

The Wingate anaerobic test was performed on the Wattbike cycle ergometer (Wattbike Ltd., Nottingham, UK), which is a combination of an air and a magnetically resisted cycle ergometer [25]. Prior to the test, a 10 min warm-up was performed consisting of constant pedaling at low intensity interspersed with brief sprints of 2–3 s. The warm-up was performed on the cycle ergometer to promote specific physiological and motor adaptation. Following the command of “start”, the subject pedaled as fast as possible without any resistance sitting on the bike. Then, 3 and 4 s after reaching the maximum speed, a load equivalent to 7.5% of body weight in kg was applied and the participant was encouraged to maintain pedaling for 30 s. Encouragement was given throughout the test to maintain the same intensity until the end of the test [25]. The fatigue index (FI) was determined by taking the percentage difference between the maximum and minimum anaerobic performance over 30 s [26].

### 2.8. RM Test Bench Press

Prior to testing, subjects warmed up on a stationary bicycle for 5 min at 75 W. Afterwards, subjects performed dynamic upper-body movements and a warm-up session at estimated intensities of 50% and 85% 1 RM for 5–10 repetitions for all exercises. The estimation of 1 RM was performed automatically with the encoder during the approach protocol. Then, the load was increased within 4–5 trials separated by at least 3 min until the 1 RM was obtained [27]. The 1 RM was stablished as the greatest weight that can be lifted once while maintaining an acceptable exercise technique.

### 2.9. WOD Assessment

After a general warm-up of 5 min on a stationary bike at low intensity and 5 min of joint mobility, a specific warm-up of 5 pull-ups, 10 push-ups, and 10 squats was performed. Next, the WOD test was performed to compare the effects of the two experimental conditions (SG vs. PG). For this, the Cindy WOD was chosen, which consists of the following tests: 20 min AMRAP (as many reps as possible), that is, they had to perform as many repetitions as possible during twenty minutes of the following exercises; 5 pull-ups/10 push-ups/15 squats. The repetitions of each exercise were counted by CrossFit-certified judges with at least two years of previous experience. The intensity of the WOD was also controlled by evaluating the mean and maximum heart rate (HR) using a Polar Vantage M heart rate monitor (Norway), as well as the subjective perceived exertion (RPE) using the Borg scale [28]. The WOD protocol was designed according to the guidelines established by Glassman (1). According to its physiological response, the Cindy WOD is described as high-intensity (HRmean = 90–95% HRmax; lactate > 14 mmol^−1^; RPE > 8) [29].

### 2.10. Blood Lactate Assessment

Capillary blood samples were collected after expelling the first drop of blood through a finger prick on the medial side of the tip of the index finger using a disposable hypodermic lancet (Accu-Chek Safe-T-Pro Uno, Roche^®^, Hawthorne, CA, USA). Blood lactate concentration was measured by photometric reflectance on a validated portable lactate analyzer (Accusport, Boehringer Mannheim—Roche^®^, Hawthorne, CA, USA). Before testing, the lactate analyzer was calibrated with different standard solutions of known lactate concentrations (2, 4, 8, and 10 mmol L^−1^). Blood lactate concentrations were measured at the following times: one minute before each WOD and one minute after each WOD.

### 2.11. Blood Glucose Assessment

Capillary blood samples were collected after expelling the first drop of blood through a finger prick on the medial side of the tip of the middle finger using a disposable hypodermic lancet (Accu-Chek Safe-T-Pro Uno, Roche, Hawthorne, CA, USA). Blood glucose concentration was measured by photometric reflectance on a validated portable glucose analyzer (Lactate Pro 2, Arkray, Edina, MN, USA). All blood glucose measurements were taken from the fingertips of the left hand at the following times: one minute before and after each WOD.

### 2.12. Statistical Analysis

The data were processed using IBM SPSS 22.0 Statistics (IBM Corp., Armonk, NY, USA). A descriptive analysis was performed to show the means and standard deviations. The normality distribution of the variables was analyzed using the Shapiro–Wilk test and the homogeneity of variances using the Levene test. The Student’s *t*-test for related samples was performed on the anthropometric, body composition, and nutritional intake parameters. For the rest of the parameters, a two-way ANOVA (group effect vs. time effect) was used to show the differences between the variables. The effect size (ES) of the intervention was calculated using Cohen’s guidelines. Threshold values for ES were >0.2 (small), >0.6 (moderate), >1.2 (large), and >2.0 (very large). The percentage of change was determined. The *p* < 0.05 differences were considered statistically significant.

## 3. Results

The results obtained in this study are shown below. Table 1 shows the participants’ characteristics.

Table 2 shows the nutritional intake of participants the day prior to evaluations. No significant differences were reported between days.

Table 3 shows the repetitions of the exercises performed in the WOD (pull-ups, push-ups, and squat), heart rate, and perceived exertion. No significant differences were observed in any of the parameters analyzed. The PG performed more repetitions in all exercises. Additionally, the mean and maximum HR of the PG was higher than that of the SG.

Figure 3 shows the results obtained in the vertical jump and handgrip tests. Figure 3C shows differences between groups in power values (*p* < 0.05). However, there were no significant differences in pre vs. post (time effect). In the handgrip test, SG decreased performance after performing the two WODs.

Table 4 shows the maximum, average power, and FI obtained in the Wingate test before and after performing both WODs. Similar to the previous table, no significant differences were reported. Average power was higher in SG. However, IF was slightly higher in PG.

Figure 4 shows the data obtained on the execution speed during the bench press. There were significant differences in SG in set 1 and set 2 pre-WOD, as well as in PG post-WOD (*p* < 0.05). On the other hand, there were significant differences throughout the repetitions in each series (*p* < 0.001) in both conditions, decreasing execution speed during the series. However, no differences were found when comparing the set with repetition in any conditions.

Figure 5 shows the results for the Wingate test. Differences were reported throughout the test (every 5 s) before and after performing both WODs.

Both groups showed a significant decrease throughout each test, but no significant differences were observed between groups or between the pre (A) and post (B) WOD tests.

Finally, Table 5 shows the results obtained in lactate and glucose concentrations before and after each WOD. There were significant differences in lactate and glucose concentrations before and after each WOD (*p* < 0.05). There were no significant differences between groups.

## 4. Discussion

The purpose of this study was to evaluate the effect of ingesting 30 g of HBCD intra-exercise on performance in CrossFit tests, as well as on some performance parameters directly related to the sport. It should be mentioned that both groups followed the same nutritional principles on the days prior to the evaluations to ensure similar glycogen and energy levels in both groups. In this sense, the pre-evaluation diet was designed and prescribed by a nutritionist and was controlled to ensure that the participants met these requirements. Therefore, a priori, glycogen levels should have been the same in both groups and, therefore, should not have contaminated the results.

In this sense, it has been shown that low-osmolarity CHO, such as HBCD, generates a favorable glycemic response after ingestion [30], with considerable increases in blood glucose levels observed 15 min after ingestion [31]. It has been widely reported in the literature that increased blood glucose levels can improve athletic performance through a variety of mechanisms [6,32], including preserving muscle glycogen levels for longer. This response has been observed alongside increased lactic acid production in anaerobic or high-intensity workouts [33]. In the present study, the intake of 30 g of HBCD did not affect blood glucose levels differently from the intake of the placebo, generating slightly higher post-WOD levels in the PG, but without reaching statistical significance. Likewise, no differences were observed between groups in lactate levels, with both groups presenting very similar values both at the beginning and at the end of the WODs.

It should be considered that, in the present study, the intake of HBCD was intra-exercise, after completing the first WOD. In this sense, it has been reported that once physical activity has begun, glycemic responses after CHO intakes can be slowed down, or even be non-existent if the amounts ingested do not exceed minimums [34]. In this sense, it has been reported in the literature that, to obtain a significant metabolic effect, the quantities must be higher than 45 g [35]. The amount ingested by our participants was 30 g, which is approximately 30% lower than the recommended amount and under conditions where the activity had already begun, so this may be the reason for the observed results.

In relation to the effect on performance in CF, it has been widely reported that the intake of CHO both before and during training can have beneficial effects on interval and resistance work, although these responses are not always observed [34]. In the specific case of CF, it has been observed that athletes frequently and variably consume nutritional supplements [7], and considering that it is a type of concurrent and interval sport [5], the use of CHO to improve performance follows the same principles as in these modalities. In this sense, the intake of CHO in interval modalities seems to have a clear effect on physical performance [6], having been observed that these modalities generate high rates of glycogen depletion [36]. In these modalities, a preservation of glycogen through the intake of CHO is going to have a greater impact on performance [6].

It is interesting, however, that bodybuilding work, despite having an interval character, does not seem to have the same relationship with CHO intake, with it being observed in most studies that high CHO intakes both before and during exercise do not present advantages in the medium- and short-term with respect to normal- or even low-CHO diets [34]. In the specific case of CF, it has been observed that in the short–medium term, high-CHO intakes did not present any advantage compared to others where the intake was moderate–low (<6 g/kg/day) [12]. In this context, the benefits of CHO intake are observed mainly in high-volume training, with a duration of more than 45 min and a high number of sets and repetitions [33].

The main training system in CF are the WODs. These training sessions usually have a marked metabolic character together with a high work density, with high energy demands per unit of time, but with a moderate duration (10–20 min) [37]. In the present work, they had a maximum duration of 20 min, where they had to perform the maximum number of repetitions (AMRAP). No effects were observed with HBCD supplementation on performance (number of repetitions), nor on associated physiological markers, such as the Borg scale or the average heart rate. These results may be in line with the vast majority of the literature and can be explained by several reasons. The first, and as mentioned above in the discussion, is the amount of supplement ingested, which was significantly lower than the minimum recommended amount of CHO (30 vs. 45 g) [35].

Another possible explanation has to do with hydration, as it has been reported that the intake of water alone can have a positive effect on the preservation of glycogen levels [35]. In high-intensity and intermittent efforts, the preservation of glycogen levels can significantly affect performance, and it has been shown that water intake decreases the degree of glycogen degradation by decreasing body temperature and/or the production of catecholamines [38]. In this sense, it has been reported that the intake of water alone can exert an effect similar to water with CHO dissolved in certain situations [35]. However, in the present study, no differences were observed between the pre- and post-evaluation for each group, so the moderate intake of water (375 mL in both groups) did not seem to have induced any response.

In addition to performance in specific CF tests, the effect of HBCD intake has also been evaluated on several metabolically related tests (CMJ, Handgrip, Wingate, and 1 RM in Bench Press). HBCD supplementation did not produce effects on performance in the CMJ or the handgrip test. With respect to the Wingate, although the SG group presented slightly higher values in the average power, no significant differences were obtained. Additionally, both groups presented a very similar fatigue curve, with no differences between groups or intragroup differences between the pre and post WOD evaluations.

In the bench press power test, during the pre-WOD evaluations, both groups presented very similar values. However, in the post-WOD evaluations, the PG group presented higher values of execution speed than the SG group. However, in the last series, the PG group presented a performance drop (red line) compared to the post-WOD series 1, which was significantly higher than the SG.

When interpreting these results, it is important to consider that people who are adapted to interval training have interesting adaptations at the level of the Cori cycle, which could explain certain results obtained [39]. It is noteworthy that despite starting with very similar initial levels of strength, the fact that the SG group decreased their results after the CF training (series 1) could be explained by greater fatigue derived from an acceleration of the glycolytic metabolism, as has been observed in some cases [40,41] Although the number of repetitions is reduced in the bench press test (six repetitions), it has been proven that glycogen levels can severely affect performance in this type of task. Despite having obtained lower levels, the difference between series 1 and 2 post-WOD was lower in the SG group, indicating less fatigue compared to the PG group. Although, hypothetically, an acceleration of glycolytic processes in the SG throughout the workout could have stimulated the Cori cycle, especially in the final part of the workout, this could also explain the lower level of fatigue in the SG.

In any case, the data obtained are not conclusive, and, in general, the results seem to indicate that this type of CF task (WOD), despite being performed at high intensity, does not represent a sufficient physiological demand to obtain benefits with HBCD supplementation, especially in adapted people with minimally acceptable nutrition and hydration. In this sense, WOD would be demanding to the body in a similar way to resistance training sessions, where it has already been widely proven that CHO supplementation does not generate benefits in most cases [34]. The present study is not free of limitations: (i) the sample size; (ii) supplementation was not prescribed by kilograms of weight; and (iii) only men participated. However, further research is needed to overcome the limitations found, as well as to obtain more consistent results.

## 5. Conclusions

Considering the results obtained, it can be concluded that the intake of 30 g of HBCD does not generate any effects on performance in CF tasks or on associated metabolic or physiological variables.

It can be concluded that 30 g is an insufficient amount to observe net effects on performance in CF tests and, although some differences between groups were observed in CMJ and the bench press power test, the data are not conclusive and further research is needed in this regard.

## Figures and Tables

**Figure 1 jfmk-09-00027-f001:**
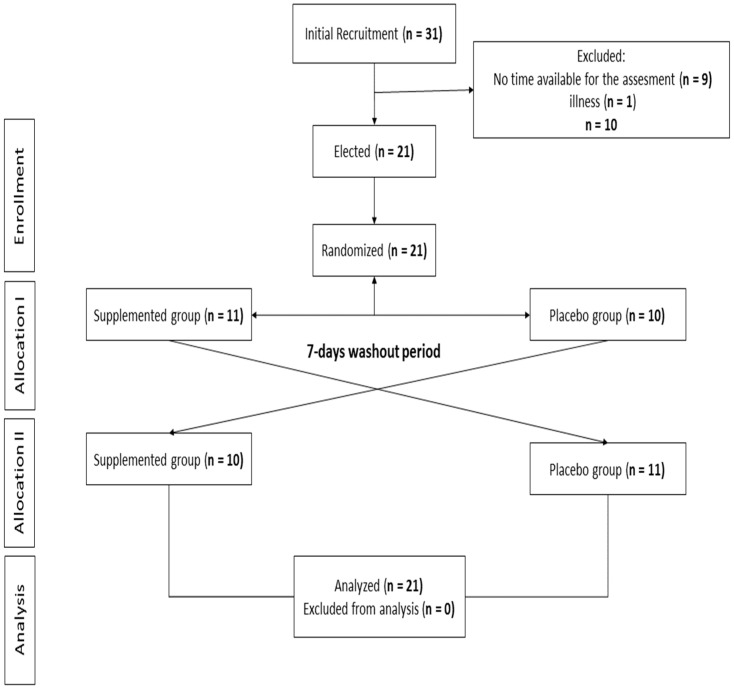
Study design flow-chart.

**Figure 2 jfmk-09-00027-f002:**
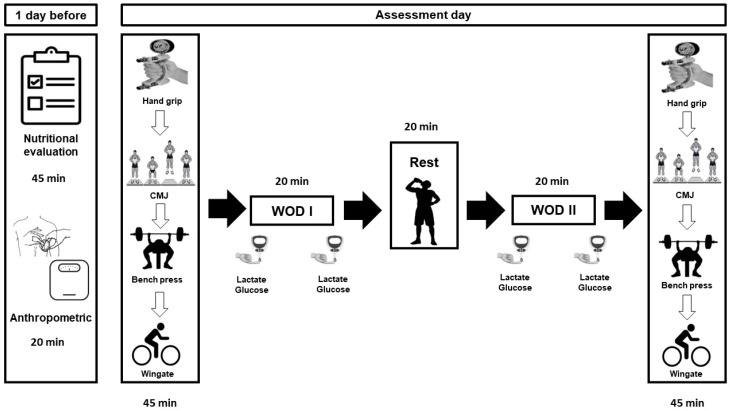
Schedule of assessments performed in the study.

**Figure 3 jfmk-09-00027-f003:**
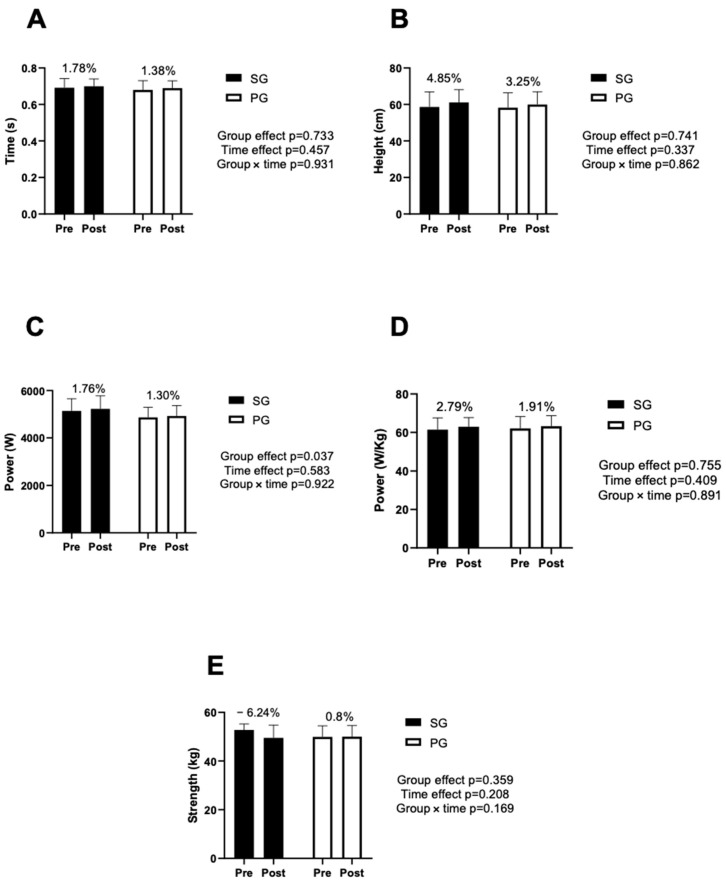
Results obtained in CMJ and handgrip. SG: supplementation group; PG: placebo group (**A**) flight time for CMJ; (**B**) height achieved for CMJ; (**C**) watts obtained for CMJ; (**D**) watts per body weight achieved for CMJ; and (**E**) results obtained for handgrip strength.

**Figure 4 jfmk-09-00027-f004:**
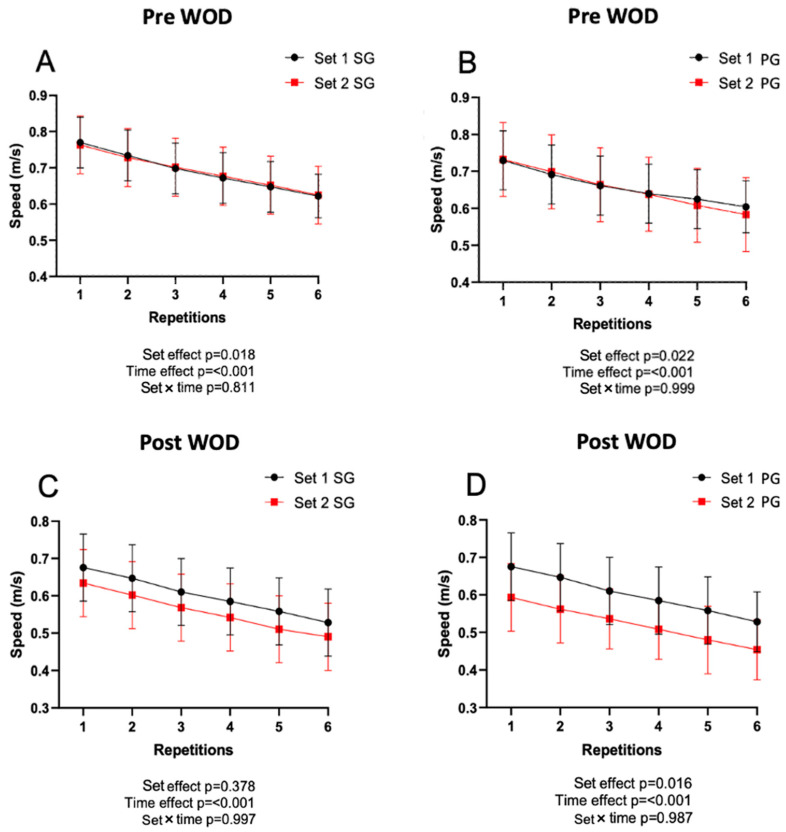
Results obtained in the bench press execution speed test. SG: supplemented group; PG: placebo group. All graphs show the speed of execution in series 1 and 2 evaluated at different times: (**A**) SG before the WOD; (**B**) PG before the WOD; (**C**) SG after the WOD; and (**D**) PG after the WOD. In addition, the statistical values are shown for series 1 and 2 both before and after the WOD.

**Figure 5 jfmk-09-00027-f005:**
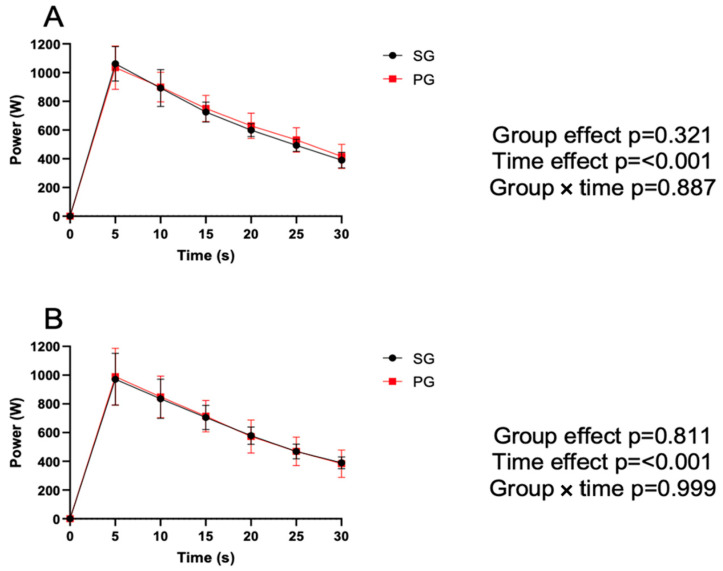
Variation in watts obtained every 5 s throughout the Wingate test in both groups before and after the WOD. SG: supplemented group; PG: placebo group; (**A**) SG and PG data before the WOD; and (**B**) SG and PG data after the WOD. Additionally, the results of the statistical significance for both evaluation times are shown.

**Table 1 jfmk-09-00027-t001:** Participants and training characteristics.

Parameters	Day 1	Day 2
Age (years)	29.5 ± 4.3	
Height (m)	1.74 ± 0.06	
Weight (kg)	72.81 ± 12.85	70.31 ± 9.53
Body mass Index	24.10 ± 2.01	23.28 ± 1.79
Muscle mass (%)	53.14 ± 1.84	52.66 ± 1.50
Lean Body Mass (kg)	56.89 ± 5.04	55.87 ± 4.73
Fat mass (%)	11.54 ± 3.21	10.92 ± 2.65
CrossFit training experience (years)	3.41 ± 1.21	
CrossFit training (weekly/hours)	4.8 ± 0.6	

**Table 2 jfmk-09-00027-t002:** Nutritional intake.

Parameters	Day 1	Day 2
Energy (kcal/d)	30.83 ± 2.47	33.57 ± 4.40
Carbohydrates (g/kg)	3.8 ± 0.8	4.1 ± 1.1
Proteins (g/kg)	1.7 ± 0.2	1.8 ± 0.4
Fat (g/kg)	1.1 ± 0.4	1.0 ± 0.3

**Table 3 jfmk-09-00027-t003:** Repetitions in WODs before and after supplementation.

	SG				PG				Group Effect	Time Effect	Group × Time
	Pre (WOD I)	Post (WOD II)	ES	Δ%	Pre (WOD I)	Post (WOD II)	ES	Δ%			
Pull-ups (rep)	100.33 ± 18.78	98.21 ± 20.41	0.11	−0.02	104.86 ± 13.13	99.45 ± 10.45	0.46	−0.05	0.521	0.378	0.691
Push-ups (rep)	195.17 ± 38.24	188.61 ± 41.67	0.16	−0.03	205.43 ± 27.15	195.78 ± 30.14	0.34	−0.04	0.356	0.219	0.722
Squat (rep)	282.42 ± 68.25	275.39 ± 70.90	0.10	−0.02	305.64 ± 42.03	292.51 ± 35.81	0.34	−0.04	0.249	0.418	0.812
HR mean (bpm)	154.0 ± 8.5	160.1 ± 10.4	−0.65	0.03	167.0 ± 4.6	173.4 ± 5.1	−1.24	0.03	0.178	0.317	0.473
HR max (bpm)	174.5 ± 7.7	178 ± 6.4	−0.50	0.02	179.6 ± 6.6	181.7 ± 4.9	−0.37	0.01	0.483	0.681	0.702
RPE (AU)	9.1 ± 0.4	9.6 ± 0.6	−1.00	0.05	9.3 ± 0.2	9.8 ± 0.3	−2.00	0.05	0.841	0.699	0.529

SG: supplementation group; PG: placebo group; AU: arbitrary units; ES: effect size; Δ%: percentage of change.

**Table 4 jfmk-09-00027-t004:** Data obtained in the Wingate Anaerobic Test.

	SG				PG				Group Effect	Time Effect	Group × Time
	Pre	Post	ES	Δ%	Pre	Post	ES	Δ%			
Mean Power (W)	681.2 ± 69.7	669.7 ± 93.5	0.14	−0.01	653.12 ± 114.4	613.5 ± 94.6	0.38	−0.06	0.241	0.535	0.364
Max Power (W)	1034.2 ± 185.4	1109.6 ± 281.9	−0.32	0.07	1030.9 ± 230.0	1115.5 ± 185.4	−0.41	0.08	0.989	0.391	0.960
FI (%)	56.4 ± 10.8	59.0 ± 12.3	−0.23	0.04	60.6 ± 14.6	63.0 ± 5.79	−0.24	0.05	0.456	0.418	0.656

SG: supplementation group; PG: placebo group; AU: arbitrary units; ES: effect size; Δ%: percentage of change.

**Table 5 jfmk-09-00027-t005:** Results obtained in glucose and lactate test.

	SG				PG				Group Effect	Time Effect	Group × Time
	Pre	Post	ES	Δ%	Pre	Post	ES	Δ%			
Lactate WOD I (mmol/L)	2.5 ± 0.7	13.3 ± 2.6	−6.55	4.32	2.2 ± 0.4	14.4 ± 3.1	5.45	−0.01	0.538	<0.001	0.416
Lactate WOD II (mmol/L)	3.1 ± 0.8	14.8 ± 3.1	−6.00	3.77	2.9 ± 0.5	13.7 ± 2.6	3.72	0.07	0.638	<0.001	0.582
Glucose WOD I (mg/dL)	96.9 ± 19.0	112.7 ± 14.3	−0.97	0.16	98.1 ± 20.5	116.5 ± 18.3	−0.95	0.18	0.792	0.032	0.610
Glucose WOD II (mg/dL)	103.0 ± 36.6	116.4 ± 11.6	−0.56	0.13	95.1 ± 9.5	129.83 ± 21.0	−2.28	0.36	0.407	0.011	0.536

SG: supplementation group; PG: placebo group; AU: arbitrary units; ES: effect size; Δ%: percentage of change.

## Data Availability

The data presented in this study are available on request from the corresponding author.

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
