# Peer review of "Acute Effects of 30 g Cyclodextrin Intake during CrossFit® Training on Performance and Fatigue"

_jfmk, 2024, doi:10.3390/jfmk9010027_

Round 1
Reviewer 1 Report
Comments and Suggestions for Authors
General
The authors have sought to investigate the short term effects of a novel CHO supplementation on Cross-Fit performance. The study has used an appropriate design but the volume of supplementation needs to be justified and further consideration given to providing the supplementation volume based on body weight.
Specific
Ln 81; Did you mean to refer to cyclodextrin and not maltodextrine in this sentence?
Ln 84; No you are referring to Clusterdextrine, so is this a commercial trademarked product or how is this different to the CHO products already introduced?
Ln 88; Please justify your sample size, and define the variables and values used to calculate the required sample.
Ln 123-125; Can the authors please include measures of test-retest reliability for the four tests in this cohort, and if possible an indication of the smallest worthwhile change in each.
Ln 129; Was the amount consumed per kg of body weight or an absolute amount regardless of the weight of the participant? If not normalised has the impact of this been discussed?
Ln 158; Use of the terms, "On the other hand" is very passive writing consider changing to, 'Alternatively' or 'In contrast,'
Ln 193; Yes the CMJ is highly reliable but the extent of variability does change depending on the cohort performing the assessment and the device used to measure the performance. Please refer to earlier comment. The jump mat system measures flight time and then calculates the jump height based on the participants body weight.
Ln 206; Was the participant seated or standing for the duration of the test?
Ln 258; I strongly recommend that the authors include effect sizes and confidence intervals to assist the reader to interpret the statistical outcomes.
Figure 3; I suggest the authors use a figure style that better represents the spread of the data such as a box and whisker plot or a stacked dot plot. There is no need for figure titles as there are identified by the panel letter. However the Y axis does need a title and the unit of measurement.
Ln 274; CMJ flight time and jump height are directly related however there is substantially more variation in the jump height reported, why?
Figure 4 & 5; The Y axis is not labelled, a unit of measure is given but not an axis title
Ln315-322; I do not see the purpose of this information, it has been adequately described in the methods and does not need to be repeated unless there is a specific discussion point that needs to refer to this information.
Ln 383; Please justify the choice to only supplement with 30g if the minimum recommended dose was 45g. Furthermore as stated what is the potential impact of normalising the does to kg body weight?
Author Response
Firstly, we are very grateful for the considerations of the comments done regarding the first version of the paper. The revised article has improved in some points. The changes have been highlighted in red colour in the revised manuscript.
General
The authors have sought to investigate the short term effects of a novel CHO supplementation on Cross-Fit performance. The study has used an appropriate design but the volume of supplementation needs to be justified and further consideration given to providing the supplementation volume based on body weight.
Authors: Thank you for your comment. It is one of the first studies to analyze the intrasession effect of CHO intake. Therefore, we believe that the design can be improved. However, this research could begin a path for future researchers to analyze the effect of different quantities and supplements in the world of Crossfit. This has been added as a limitation.
Specific
Ln 81; Did you mean to refer to cyclodextrin and not maltodextrine in this sentence?
Authors: Yes. It has been modified.
Ln 84; No you are referring to Clusterdextrine, so is this a commercial trademarked product or how is this different to the CHO products already introduced?
Authors: It has been modified. It is cyclodextrin.
Ln 88; Please justify your sample size, and define the variables and values used to calculate the required sample.
Authors: Statistical power has not been calculated and has been established as a limitation. In this sense, CrossFit is an emerging modality in our region and we only had access to a single sports center where this modality is carried out.
Ln 123-125; Can the authors please include measures of test-retest reliability for the four tests in this cohort, and if possible an indication of the smallest worthwhile change in each.
Authors: Thanks for your appreciation. In this sense, test-retest reliability tests of the tests used were not carried out since all have been previously used in the scientific literature and are validated. By not introducing any new, non-validated test, we did not believe it was appropriate to perform it. If you consider it so, we can include it within the limitations of our study.
Ln 129; Was the amount consumed per kg of body weight or an absolute amount regardless of the weight of the participant? If not normalised has the impact of this been discussed?
Authors: Thanks for your question. As we mentioned in the last question, we decided to administer it in absolute value so that everyone ingested the same amount and to be able to control the possible stomach upsets it causes. In this sense, it is true that other substances such as caffeine can be administered relative to weight, but we did not decide to do so in the present study.
Ln 158; Use of the terms, "On the other hand" is very passive writing consider changing to, 'Alternatively' or 'In contrast,'
Authors: It has been modified.
Ln 193; Yes the CMJ is highly reliable but the extent of variability does change depending on the cohort performing the assessment and the device used to measure the performance. Please refer to earlier comment. The jump mat system measures flight time and then calculates the jump height based on the participants body weight.
Authors: We agree with you, however we did not have the gold standard to measure that variable, which would be a force platform, therefore we had to carry it out with the jump mat system.
Ln 206; Was the participant seated or standing for the duration of the test?
Authors: The participant was sitting on the bike.
Ln 258; I strongly recommend that the authors include effect sizes and confidence intervals to assist the reader to interpret the statistical outcomes.
Authors: According to your consideration, we have included the effect size and the percentage of change for each of the variables.
Figure 3; I suggest the authors use a figure style that better represents the spread of the data such as a box and whisker plot or a stacked dot plot. There is no need for figure titles as there are identified by the panel letter. However the Y axis does need a title and the unit of measurement.
Authors: Thanks for your appreciation. The variable and its unit of measurement have been added to the Y axis. In addition, Figure 4 has been reorganized for better visualization and the repetitive figure titles have been removed.
Ln 274; CMJ flight time and jump height are directly related however there is substantially more variation in the jump height reported, why?
Authors: Thanks for your appreciation. Jump height and flight time in a vertical jump are related measurements, but several biomechanical, physiological, or even muscular factors can contribute to variability in jump height between different individuals, even if flight time is similar. Therefore, we understand that this greater % change in jump height is due to differences between subjects.
Figure 4 & 5; The Y axis is not labelled, a unit of measure is given but not an axis title
Authors: It has been modified.
Ln315-322; I do not see the purpose of this information, it has been adequately described in the methods and does not need to be repeated unless there is a specific discussion point that needs to refer to this information.
Authors: It has been removed in the text.
Ln 383; Please justify the choice to only supplement with 30g if the minimum recommended dose was 45g. Furthermore as stated what is the potential impact of normalising the does to kg body weight?
Authors: Thanks for your question. As we mentioned above, we decided to give that amount because a higher amount could cause stomach discomfort to the participants and prevent them from carrying out the exercise protocol. In this sense, we prioritized that the subjects finish the protocol rather than increase the administered dose.
Reviewer 2 Report
Comments and Suggestions for Authors
Grijota et al investigated here the effects of carbohydrate ingestion on the performance and blood biomarkers of CrossFit athletes during two consecutive 20 min-physical challenges (WODs).
The objective of the study is clear and the authors meticulously described the experimental methods applied herewith. These are positive appreciations of this MS. However, I still have some questions and suggestions to the authors that should be answered before the acceptance of this MS, as presented below:
(i) Abstract: the acronyms WODs and HBCD were not previously described in the text and, therefore, will confuse the Abstract readers. Please correct that.
(ii) I presume the authors should reinforce the information about the carbohydrate supplement used (HBCD), since it is not an usual compound normally used in sports, exercises (such as maltodextrin). In time, my suggestion is that the whole information presented from lines 323 - 334 should be repositioned in the Introduction section for that purpose.
(iii) Please clarify the information presented in Table 1: at that point (of reading), readers do not have any idea what "Day 1" and "Day 2" mean. Or simply reposition TAble 1 further in the text.
(iv) Figure 2 should be rebuild. If we consider the factor "TIME" as columns in the figure, the first column "1 Day Before" brings down all procedures executed during that day. Therefore, the second column "Assessment Day" should also present, WITHIN, columns representing the time course of events executed during that day. For example: "Handgrip-CMJ-BenchPress-Wingate" should compose one narrow column within "Assessment Day" juxtaposed to the "1 Day before" column. At the end, we should observe SIX columns:
"1 day before"
"Handgrip-CMJ-BenchPress-Wingate"
"WOD1"
"Rest"
"WOD2", and
"Handgrip-CMJ-BenchPress-Wingate" , the last five columns within "Assessment DAy". I hope I was clear in my suggestion.
(v) Please , use the acronym HBCD throughout the whole MS. For example, avoid "maltodextrin" in line 253
(vi) line 224. What is "AMRAP"?
(vii) Why there are no measurements of pre/post WOD1 and pre/post WOD2 in Table 3. It is confusing to compare pre-WOD1 with post-WOD2. The authors should justify why did they compared these checkpoints in their study.
(viii) I think the conclusion should avoid the discussion about the 8% dilution of the 30g HBCD, because it was not into the initial hypothesis of the study.
Comments on the Quality of English Language
Recheck the whole MS after corrections.
Author Response
Firstly, we are very grateful for the considerations of the comments done regarding the first version of the paper. The revised article has improved in some points. The changes have been highlighted in red colour in the revised manuscript.
Grijota et al investigated here the effects of carbohydrate ingestion on the performance and blood biomarkers of CrossFit athletes during two consecutive 20 min-physical challenges (WODs).
The objective of the study is clear and the authors meticulously described the experimental methods applied herewith. These are positive appreciations of this MS. However, I still have some questions and suggestions to the authors that should be answered before the acceptance of this MS, as presented below:
(i) Abstract: the acronyms WODs and HBCD were not previously described in the text and, therefore, will confuse the Abstract readers. Please correct that.
Authors: It has been modified in the text.
(ii) presume the authors should reinforce the information about the carbohydrate supplement used (HBCD), since it is not an usual compound normally used in sports, exercises (such as maltodextrin). In time, my suggestion is that the whole information presented from lines 323 - 334 should be repositioned in the Introduction section for that purpose.
Authors: It has been modified in the text.
(iii) Please clarify the information presented in Table 1: at that point (of reading), readers do not have any idea what "Day 1" and "Day 2" mean. Or simply reposition Table 1 further in the text.
Authors: It has been modified in the text.
(iv) Figure 2 should be rebuild. If we consider the factor "TIME" as columns in the figure, the first column "1 Day Before" brings down all procedures executed during that day. Therefore, the second column "Assessment Day" should also present, WITHIN, columns representing the time course of events executed during that day. For example: "Handgrip-CMJ-BenchPress-Wingate" should compose one narrow column within "Assessment Day" juxtaposed to the "1 Day before" column. At the end, we should observe SIX columns:
"1 day before"
"Handgrip-CMJ-BenchPress-Wingate"
"WOD1"
"Rest"
"WOD2", and
"Handgrip-CMJ-BenchPress-Wingate" , the last five columns within "Assessment DAy". I hope I was clear in my suggestion.
Authors: It has been modified.
(v) Please, use the acronym HBCD throughout the whole MS. For example, avoid "maltodextrin" in line 253
Authors: It has been modified.
(vi) line 224. What is "AMRAP"?
Authors: It has been modified.
(vii) Why there are no measurements of pre/post WOD1 and pre/post WOD2 in Table 3. It is confusing to compare pre-WOD1 with post-WOD2. The authors should justify why did they compared these checkpoints in their study.
Authors: Thanks for your appreciation. Intra-WOD comparisons are represented by the time effect.
(viii) I think the conclusion should avoid the discussion about the 8% dilution of the 30g HBCD, because it was not into the initial hypothesis of the study.
Authors: It has been removed.
Reviewer 3 Report
Comments and Suggestions for Authors
The main objective of this study was to investigate the influence of carbohydrate intake (Clusterdextrine) on performance during the performance of two consecutive WODs lasting 20 minutes each. This study is very great for readers. I have few comments prior the acceptance.
Abstract:
To add more numbers and details.
Introduction:
The hypothesis is no clear. Please, to add.
Methods:
To add details of lean mass and hydration status.
Results:
To add number of lean body mass and hydration.
To add the energy intake in kcal/kg/d not kcal/d.
Discussion:
To discussion better the effects of supplements on WOD impact.
Author Response
I am writing to reply point-by-point to the reviewers' comments and the editorial office's requests. Firstly, we are very grateful for the considerations of the comments done regarding the first version of the paper. The revised article has improved in some points. The changes have been highlighted in red colour in the revised manuscript.
The main objective of this study was to investigate the influence of carbohydrate intake (Clusterdextrine) on performance during the performance of two consecutive WODs lasting 20 minutes each. This study is very great for readers. I have few comments prior the acceptance.
Abstract:
To add more numbers and details.
Authors: It has been added.
Introduction:
The hypothesis is no clear. Please, to add.
Authors: It has been added.
Methods:
To add details of lean mass and hydration status.
Authors: Thanks for your appreciation. We have added the value of lean body mass, but not the hydration status since since electrical bioimpedance was not carried out, we do not have that data.
Results:
To add number of lean body mass and hydration.
Authors: We have added the value of lean body mass in table 1.
To add the energy intake in kcal/kg/d not kcal/d.
Authors: It has been modified.
Discussion:
To discussion better the effects of supplements on WOD impact.
Authors: Thanks for your consideration. The effects of supplements on WOD impact have been discussed on line 405-414.
Round 2
Reviewer 2 Report
Comments and Suggestions for Authors
From my point of view, the authors properly corrected their MS in agreement with my suggestions. I consider it ready for publication.
Author Response
Dear reviewer, thank you very much for your comments and helping us improve the manuscript.